# It Takes Four to Tango: Multiagent Selfplay for Automatic Curriculum Generation

**Yuqing Du**
UC Berkeley
yuqing_du@berkeley.edu

**Pieter Abbeel**
UC Berkeley
pabbeel@berkeley.edu

**Aditya Grover**
UCLA
adityag@cs.ucla.edu

## Abstract

We are interested in training general-purpose reinforcement learning agents that can solve a wide variety of goals. Training such agents efficiently requires automatic generation of a *goal curriculum*. This is challenging as it requires (a) exploring goals of increasing difficulty, while ensuring that the agent (b) is exposed to a diverse set of goals in a sample efficient manner and (c) does not catastrophically forget previously solved goals. We propose *Curriculum Self Play* (CuSP), an automated goal generation framework that seeks to satisfy these desiderata by virtue of a multi-player game with 4 agents. We extend the asymmetric curricula learning in PAIRED (Dennis et al., 2020) to a symmetrized game that carefully balances cooperation and competition between two off-policy student learners and two regret-maximizing teachers. CuSP additionally introduces entropic goal coverage and accounts for the non-stationary nature of the students, allowing us to automatically induce a curriculum that balances progressive exploration with anti-catastrophic exploitation. We demonstrate that our method succeeds at generating an effective curricula of goals for a range of control tasks, outperforming other methods at zero-shot test-time generalization to novel out-of-distribution goals.

## 1 Introduction

Reinforcement learning (RL) has seen great success in a range of applications, from game-playing (Silver et al., 2016; Schrittwieser et al., 2020) to real world robotics tasks (OpenAI et al., 2019). However, these accomplishments are often in the realm of mastering a single specific task. For many applications, we further desire agents who are capable of generalizing to a variety of tasks. This can achieved via goal-conditioned RL, where an RL agent is trained to solve many goals by specifying a sequence of goals, i.e. a goal curriculum. Manually designing such curricula is tedious and moreover, does not scale due to being environment specific and heuristic-driven. Even when it is easy to engineer a curriculum, we would prefer curricula that can dynamically adapt to the learner's current capabilities. For example, effective human tutors use an understanding of their pupil's current knowledge and learning trajectory to create a curriculum.

Our work builds on two broad desiderata for curriculum generation (Bengio et al., 2009) that enable sample-efficient learning of goal-conditioned policies: *progressive exploration* and *anti-catastrophic exploitation*. Progressive exploration refers to curricula where the agent gradually learns to solve tasks of increasing difficulty and diversity. Anti-catastrophic exploitation ensures that the agent periodically resolves previously attempted goals that could become harder to solve during the course of training a.k.a. catastrophic forgetting (French, 1999; Kirkpatrick et al., 2017).

Many prior works have attempted to incorporate aspects of these criteria for automatic curriculum generation. One approach is to parameterize a goal generator and pit it against a discriminator that assesses goal difficulty (Florensa et al., 2018) or feasibility (Racaniere et al., 2020) for the current agent. Racaniere et al. (2020) also address desired qualities of validity and diversity by crafting specific losses for these characteristics. However, these approaches rely on learning an accurate discriminator for assessing goal difficulty, which can be challenging. Rather than employing discriminators in the standard adversarial minimax formulation, Dennis et al. (2020) proposes an alternate minimax objective based on the agent's regret. The 'expert' used for computing the regret is another goal-conditioned policy whose utility is maximized by an environment generator.

---

Code available at https://github.com/yuqingd/cusp

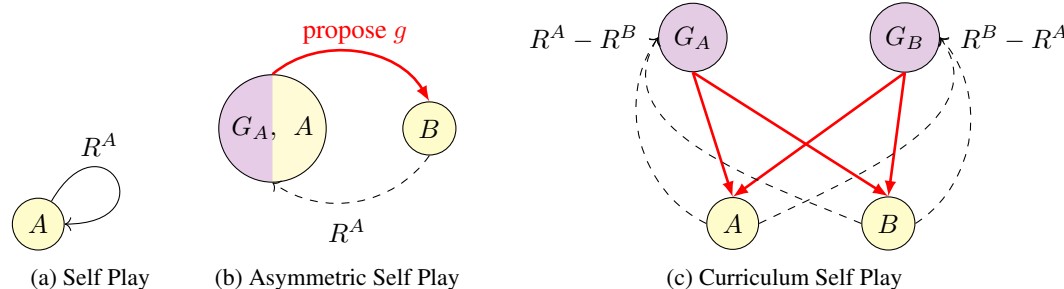

Figure 1: Interactions between learners $A$ , $B$ and teachers $G_A, G_B$ in Self Play variations. In (a), $A$ plays against variations of itself in a zero-sum game to automatically generate a curriculum. In (b), $A$ acts as both an agent and a demonstrator by being rewarded for proposing hard goals to $B$. In (c), we separate the goal generation from the demonstrators and symmetrize the whole system.

An alternative method for automatic curriculum generation that has seen great success is self-play, where an agent plays against other versions of itself (Silver et al., 2016; Bansal et al., 2018; Baker et al., 2019) (Fig. 1a). Unfortunately, this method does not apply directly to complex tasks where we do not have the structure of a two player, zero-sum game. A promising variation is Asymmetric Self-Play (ASP) (Sukhbaatar et al., 2018b), which involves a minimax game between two similar agents: a goal demonstrator, Alice, and the desired goal-conditioned agent, Bob (Fig. 1b). Bob is asked to solve or reverse tasks demonstrated by Alice in the same environment. To motivate Alice to propose successively challenging goals, Alice is rewarded whenever Bob fails. This method inherently captures an easy-to-hard curriculum, since any goal solved by Alice is achievable by Bob in principle. However, ASP tends to be sample inefficient as it is bottlenecked by Alice's ability to remember earlier goals and explore a diverse goal space.

Our work aims to bring together the strengths of goal generative methods and ASP while addressing the two desiderata. Our work focuses on the multi-goal setting with the curriculum generator controlling goals Portelas et al. (2020). We propose *Curriculum Self Play* (CuSP), a symmetric multi-agent setup for automatic curricula generation. CuSP optimizes for two off-policy goal-conditioned students, Alice and Bob, and two regret-maximizing goal generators, $G_A$ and $G_B$ (Fig. 1c). Drawing inspiration from the minimax regret objective in PAIRED (Dennis et al., 2020), we designate each goal generator as being 'friendly' to one of the students ($G_A$ for Alice, $G_B$ for Bob), and optimize each goal generator using the difference in utility between the preferred student and the other student (i.e., $G_A$ is rewarded $R^A - R^B$, where $R^A, R^B$ are Alice's and Bob's returns respectively when challenged with the same goal). This encourages proposing goals that are neither too easy nor too hard. With our symmetrized setup, Alice and Bob each have a 'friendly' goal generator that is inclined to propose more feasible goals without being too simple, and an 'unfriendly' goal generator that is inclined to propose more challenging goals. As the tasks increase in difficulty overall, the former's cooperation helps in scaffolding learning while the latter's competition helps push the boundaries of the agent's capabilities.

We further address goal diversity and sample efficiency by leveraging the off policy entropy-regularized optimization of Soft Actor Critic (SAC) (Haarnoja et al., 2018) for training the goal generators. The goal generators' replay buffers track the regrets for all proposed goals. To account for non-stationary learners, we propose an update rule for the regrets in the goal generators' replay buffers based on the critic networks of Alice and Bob. In doing so, we constantly keep track of the learners' current abilities and can propose increasingly challenging goals. At the same time, using the regret objective also places the current abilities in context. For example, if Bob has forgotten how to achieve a particular goal, the regret $R^A - R^B$ increases and $G_A$ is more inclined to re-propose that goal. Thus the combination of using a replay buffer and the regret objective aids in addressing the twin goals of progressive exploration and anti-catastrophic exploitation.

Compared to prior methods for goal generation, we demonstrate that CuSP is able to satisfy the desiderata for good curricula. We show that this translates to improved sample efficiency and generalization to novel out-of-distribution goals across a range of robotics tasks, encompassing navigation, manipulation, and locomotion. We highlight some emergent skills that arise via CuSP when generalizing to particularly challenging goals, and find that our method is able to handle cases when portions of the goal space are infeasible.

## 2 BACKGROUND

**Problem Setup.** We model the environment and the goal-conditioned learner using a Markov decision process, $\mathcal{M} = \langle \mathcal{S}, \mathcal{A}, \mathcal{T}, \mathcal{R}, \mathcal{G}, \gamma \rangle$, where $\mathcal{S}$ is the state space, $\mathcal{A}$ is the action space, $\mathcal{T} : \mathcal{S} \times \mathcal{A} \times \mathcal{S} \to \mathbb{R}$ is the environment transition function, $\mathcal{R} : \mathcal{G} \times \mathcal{S} \to \mathbb{R}$ is the goal-conditioned reward function, $\mathcal{G} \subseteq \mathcal{S}$ is the goal space, and $\gamma$ is a discount factor. We consider both dense and sparse reward functions: $r(s|g) = -d(s, g)$, where $d$ is a distance metric (e.g. $L_2$ norm) between the state $s$ and the goal $g$ for the former, or $r(s|g) = \mathbb{1}\{d(s, g) < \epsilon\}$ for the latter. Our objective is to learn a policy $\pi(a_t|s_t, g)$ that maximizes the success rate of reaching all goals $g \in \mathcal{G}$. To assess ability to generalize to out-of-distribution goals, we split the goal space into $\mathcal{G}_{id}$ and $\mathcal{G}_{ood}$ where the goal generators can only propose goals $g \in \mathcal{G}_{id}$, and we evaluate on sampled $g \in \mathcal{G}_{ood}$.

**Self Play (SP).** SP is an approach for automatic curriculum generation in competitive, zero-sum games. In this paradigm, an agent plays against past versions of itself in order to automatically generate a learning curriculum (Figure 1a). That is, we consider Alice and Bob as two versions of the same underlying agent, and each agent's objective is the negative of the others', so $R^A = -R^B$. However, this approach is only applicable for symmetric zero-sum games, and does not lend itself to learning curricula for goal-conditioned agents in other setups.

**Asymmetric Self Play (ASP).** ASP (Sukhbaatar et al., 2018b) extends self play to non-zero sum games by proposing an asymmetric parametrization of two competing agents. During each episode, Alice's objective is to propose and demonstrate a task for Bob to complete (Figure 1b). In the goal-conditioned paradigm, we take Alice's final state $s_T^A$ as the proposed goal $g$. Alice and Bob now have different objectives: Bob's return, $R^B = \sum_t \gamma^t r(s_t^B|g)$, is simply based on reaching the proposed goal, while Alice's reward is based on Bob's inability to reach the goal, $R^A = \mathbb{1}(d(s_T^B, g) \geq \epsilon)$. While this approach lends itself well to more complex tasks such as robotic manipulation (OpenAI et al., 2021), it has poor sample efficiency. Since we rely on a learning agent, Alice, to propose and solve goals, the generated goals are limited by Alice's own ability to explore and learn challenging yet feasible goals. This can be particularly problematic towards the start of training, where we can waste many episodes proposing simple and redundant goals for Bob.

## 3 AUTOMATIC CURRICULUM GENERATION VIA CuSP

Our curriculum generation method consists of four main players: two goal-conditioned learning peers, Alice $\pi_A$ and Bob $\pi_B$, and two goal generators, $G_A$ and $G_B$. $\pi_A(a|s, g)$ and $\pi_B(a|s, g)$ are both optimized to maximize the discounted sum of goal-conditioned rewards $R(g) = \sum_t \gamma^t r^t(g)$, across proposed goals $g \in \mathcal{G}_{id}$. Aside from any initial randomness in initializing $\pi_A$ and $\pi_B$, the two policies are parameterized with the same architecture. $G_A(g|s_0, z)$ and $G_B(g|s_0, z)$ are parametrized as policies that output a goal $g \in \mathcal{G}_{id}$ given an initial observation of the environment $s_0$ and a latent noise variable $z \sim \mathcal{N}(0, 1)$. The inclusion of such a noise variable draws from generative models (e.g. GANs, VAEs) where such a $z$ is the input to the generator, and is also found in prior goal generative methods (Florensa et al., 2018). The goal generators are optimized to maximize the regret of their corresponding agent: $\mathfrak{R}^{G_A} = R^A(g) - R^B(g)$ and $\mathfrak{R}^{G_B} = -\mathfrak{R}^{G_A} = R^B(g) - R^A(g)$.

We refer to our overall curriculum generation approach as *Curriculum Self Play* (CuSP). We present an overview of CuSP in Algorithm 1 and illustrate it in Figure 1c. The algorithm proceeds in rounds. In every round, both goal generators propose a goal, $g_A$ from $G_A$ and $g_B$ from $G_B$. To keep the symmetrization fair and hopefully scaffold learning initially, we first rollout the 'easier' goal for each respective agent (ie. $g_A$ for Alice, $g_B$ for Bob) and compute the corresponding environment returns and update the learners. Next, we rollout the 'harder' goal for each respective agent (ie. $g_B$ for Alice, $g_A$ for Bob) and update again. At the end of the round, we update the goal generators.

**Symmetrization.** The regret objective $R^A - R^B$ helps $G_A$ propose goals that are challenging for Bob while still being feasible for Alice. However, this can be detrimental when Alice and Bob's performance diverge too quickly. To address this, we propose symmetrizing the multi-agent system by introducing a corresponding goal generator $G_B$ whose regret objective is $R^B - R^A$, essentially pitting it in a zero-sum game against $G_A$. Having these two goal generators produces two types of goals for each agent: a) a feasible yet not too simple goal from the 'friendly' goal generator ($G_A$ for Alice, $G_B$ for Bob), which can help scaffold learning quickly, and b) a challenging yet not too difficult goal from the 'unfriendly' goal generator ($G_A$ for Bob, $G_B$ for Alice), which can help push

---

**Algorithm 1:** Curriculum Self Play (CuSP)

---

Initialize goal-conditioned learners $\pi_A, \pi_B$, goal generators $G_A, G_B$;
**while** *not converged* **do**
> // generate goals
> $g_a \sim G_A, g_b \sim G_B$;
> // evaluate easier goals first
> $R^A_{Easy} \leftarrow rollout(\pi_A, g_A); R^B_{Easy} \leftarrow rollout(\pi_B, g_B)$;
> // update policies
> $\pi_A \leftarrow train\_learner(\pi_A, R^A_{Easy}); \pi_B \leftarrow train\_learner(\pi_B, R^B_{Easy})$;
> // evaluate harder goals second
> $R^A_{Hard} \leftarrow rollout(\pi_A, g_B); R^B_{Hard} \leftarrow rollout(\pi_B, g_A)$;
> // update policies
> $\pi_A \leftarrow train\_learner(\pi_A, R^A_{Hard}); \pi_B \leftarrow train\_learner(\pi_B, R^B_{Hard})$;
> // compute regrets and update generators
> $G_A \leftarrow train\_generator(G_A, \{(R^A_{Easy} - R^B_{Hard}), g_A\}, \{(R^A_{Hard} - R^B_{Easy}), g_B\})$;
> $G_B \leftarrow train\_generator(G_B, \{(R^B_{Easy} - R^A_{Hard}), g_B\}, \{(R^B_{Hard} - R^A_{Easy}), g_A\})$;

**end**

---

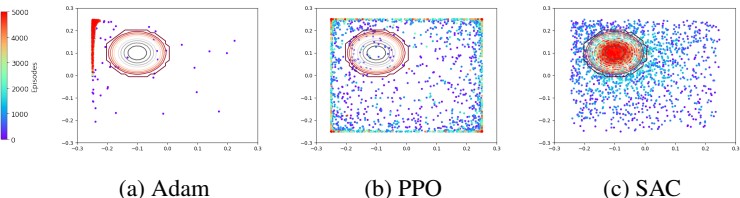

(a) Adam          (b) PPO          (c) SAC

Figure 2: Performance of different optimization methods on a toy landscape that is primarily flat except for a single gaussian centered at (-0.1, 0.1) shown by the contours. Red shows most recently proposed points. SAC is able to find the peak while the other methods get stuck in the flat regions.

the boundaries of the agent's abilities. Both of these properties help encourage the generation of goals that are **feasible** yet progressively challenging.

**Goal Generator Design.** We motivate our goal generator design with our proposed desiderata: progressive exploration with anti-catastrophic exploitation. Although the goal generator is not an RL agent in the traditional sense – it only 'acts' in a single timestep by proposing a goal and its 'observation' only changes based on a random noise variable – optimizing such an explore-exploit tradeoff effectively naturally inspires the use of an RL objective, and RL objectives have previously been used for one-step optimization (e.g., one-step Q-learning (Watkins & Dayan, 1992)).

Thus, to train the goal generators we use an RL learning objective and parameterization inspired by a single-time step variant of Soft-Actor Critic (SAC) (Haarnoja et al., 2018). SAC is an off policy algorithm that optimizes for a trade-off between expected reward and policy entropy $H(\pi)$ with a

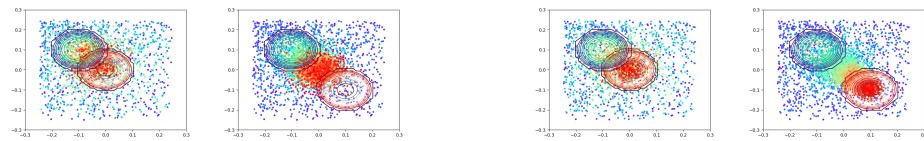

(a) SAC *without* regret updates at 2500 and 5000 steps.   (b) SAC *with* regret updates at 2500 and 5000 steps.

Figure 3: SAC optimizing a non-stationary toy landscape that is primarily flat, but has a gaussian peak starting at (-0.1, 0.1) and moving diagonally to (0.1, -0.1). Red shows most recently proposed points. Without the regret updates, the optimizer lags and cannot find the optimum.

coefficient $\alpha$. In our setup, our goal generator only acts once to propose a goal per episode, so each 'trajectory' is only a single time step consisting of the initial state $s$ (for which we concatenate the initial environment observation $s_0$ with a latent noise variable $z$), the proposed goal action $g$, and we get the corresponding "reward" for the goal generator policy by computing the regret between $\pi_A$ and $\pi_B$ on $g$. Thus the objective for each goal generator policy $G$ is given by

$$\max_G \mathbb{E}_{g \sim G} \left[ \underbrace{\mathfrak{R}^G(s, g)}_{\text{exploitation}} + \alpha \underbrace{H(G)}_{\text{exploration}} \right]$$

where the regret term addresses anti-catastrophic exploitation by prioritizing goals where regret has increased, and the entropy term addresses progressive goal space exploration.

We equip the SAC goal generators with a replay buffer to store all past goals and their regrets. This helps address **forgetting** and improve the sample efficiency of goal generation compared to prior methods which optimize the generator as another agent using on-policy algorithms such as PPO (Schulman et al., 2017). We further take advantage of SAC's entropy regularization to progressively increase the **diversity** of the goals, with consideration of Alice and Bob's capabilities using the stored regrets. To empirically motivate the use of SAC, we compare SAC against PPO and Adam (Kingma & Ba, 2015) in optimizing a synthetic landscape. One challenge of the regret objective is that at the beginning of training, the landscape will mostly be flat when $\pi_A$ and $\pi_B$ are similarly unskilled. In Fig. 2, we find that only SAC is proficient at optimizing a primarily flat landscape.

**Dynamic Regret Updates for Non-stationary Agents.** However, using an off policy RL algorithm for optimizing such a goal generator requires further modifications. For more typical agents, the received rewards are a stationary property of the environment. In our case this is no longer true – Alice and Bob are continuously learning and part of the multi-agent environment, so the regrets for the same goal will also change over time. To address this issue, we would like to update the non-stationary regret values in the goal generator's replay buffer. However, it is expensive for Alice and Bob to replay all the goals in the environment. As a sample efficient trick, we leverage the observation that the critics of Alice and Bob can provide a current estimate of their performance. Hence, after every round, we update the regrets of previous goals in the replay buffer using the difference between the corresponding value estimates from the critic networks of Alice and Bob. This allows us to keep an updated estimate of **feasibility** based on Alice and Bob's current skills.

To empirically validate that updating the replay buffer regrets helps in a non-stationary landscape, we carry out investigations of a SAC goal generator both with and without regret updates in Fig. 3. As hypothesized, updating the replay buffer allows the generator to find the moving optimum. The combination of the replay buffer and dynamic regret updates also strengthens our method by automatically prioritizing goals that have been forgotten. With the non-stationary regret updates, the goal generators always have an updated estimate of each learner's abilities. Thus if Bob performs worse on a past goal, the regret increases for $G^A$ and it is more likely to repropose it, addressing catastrophic **forgetting** for Bob. Likewise, symmetrization helps to address catastrophic **forgetting** for Alice. Without the corresponding $G_B$, if Alice performs worse at a goal, the regret $R^A - R^B$ decreases and $G_A$ is less likely to repropose it unless Bob's performance decreases even further. This can potentially lead to neglecting to propose these challenging goals for Bob.

**Summary.** Inspired by Racaniere et al. (2020), we specify desired qualities of a goal generator:

- **Progressive Feasibility** – the goals should be incrementally more difficult, accounting for the agent's current skill level and the fact that the agent is a non-stationary learner.
- **Progressive Diversity** – the goals should encourage increasing exploration of goals.
- **Anti-Forgetting** – if previous goals are forgotten, they should be reproposed.

We summarize the contributions of each component of our method to desired qualities of feasibility, diversity, and anti-forgetting in Table 1. We visualize generated goal distributions in Appendix A and ablate over each component of Table 1 in Appendix B.5.1 to illustrate their importance.

## 4 EXPERIMENTS

**Environments.** We test our method across a suite of continuous control tasks adapted from the Deepmind Control Suite (Tassa et al., 2020) and OpenAI Gym (Plappert et al., 2018), using the

Table 1: Contribution of each component of our method to the desired qualities.

| | Progressive Feasibility | Progressive Diversity | Anti-Forgetting |
|---|:---:|:---:|:---:|
| Entropy Regularization | | ✓ | |
| Regret Replay Buffer | | | ✓ |
| Dynamic Regret Updates | ✓ | | ✓ |
| Symmetrization | ✓ | | ✓ |

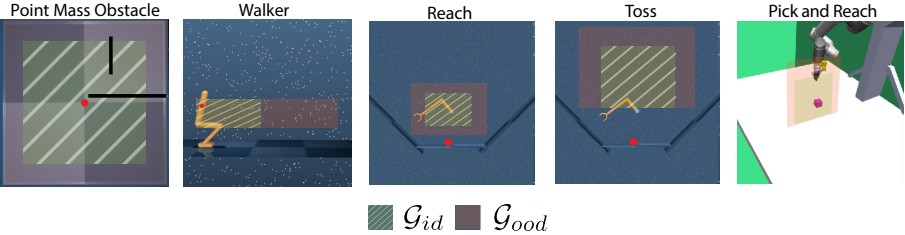

$\mathcal{G}_{id}$    $\mathcal{G}_{ood}$

Figure 4: Illustration of the five environments, $\mathcal{G}_{id}$ showing training goal space and $\mathcal{G}_{ood}$ showing test goal space. The red points indicate what is used to evaluate whether a goal was reached in each environment – the point mass coordinates for Point Mass Obstacle, the torso center location for Walker, the ball location for Reach/Toss, and the block location for Pick and Reach.

MuJoCo simulator (Todorov et al., 2012). We analyze our method across a variety of task types: *navigation*, *locomotion*, and *manipulation*. Point Mass Obstacle is a navigation task with some walls. Walker is a locomotion task with a goal torso position. Reach and Toss are manipulation tasks with a goal ball position. Toss provides an interesting challenge as gravity plays a large role – the robot must learn to aim precisely. Pick and Reach (OpenAI, 2020) is a manipulation task where success is measured by picking up a block and moving it to a target position. Environments and goal spaces are visualized in Figure 4, with details in Appendix B.2.

**Training Setup.** For CuSP, we train the goal generators using the SAC implementation from Yarats & Kostrikov (2020) with their default architecture and hyperparameters. Since symmetrization leads to similar performance from Alice and Bob in expectation, we report the results from Bob only for consistency. We compare against a domain randomization Tobin et al. (2017) baseline and two other baselines that use minimax formulations for curriculum generation: the GoalGAN approach (Florensa et al., 2018) and ASP+BC (OpenAI et al., 2021). We train all learner policies with the same architecture and SAC to isolate differences to the goal generation method.

We aim to answer the following questions with our experiments:

- Does the CuSP curricula induce agents that can generalize to novel goals from $\mathcal{G}_{ood}$?
- Does the CuSP curricula induce interesting emergent skills for solving harder goals?
- Does the CuSP curricula handle cases where portions of the goal space are poorly specified (e.g. with impossible dimensions)?

## 4.1 Test-Time Out-Of-Distribution Generalization

To assess whether our curriculum improves generalizability, we evaluate each method on its ability to reach randomly sampled out-of-distribution goals $g \sim \mathcal{G}_{ood}$. In Fig. 5 we see that across environments, generally CuSP matches or outperforms all the baselines for task success. ASP+BC's sample inefficiency due to Alice's limitations leads to poor performance, while GoalGAN does well in environments where the goal space is more symmetrical (e.g. with navigation or the reaching tasks, where most points in the goal space are similarly difficult). That said, we note GoalGAN struggles more in environments where portions of the goal space are more challenging (e.g. walking to the right or aiming and tossing upwards). We hypothesize this is due to the challenges of accurately learning the feasibility of different goals in an asymmetric goal space. Lastly, we find

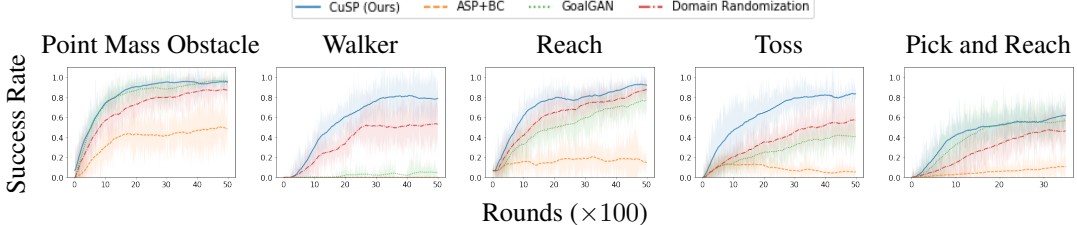

Figure 5: Success rates across environments on random goals sampled from $\mathcal{G}_{ood}$. Results averaged across 3 seeds for each method. Each round corresponds to goal generation, then rollouts, then the respective goal generator updates. Generally we find our method matches or outperforms all baselines in out-of-distribution generalization, especially when tackling the harder locomotion task.

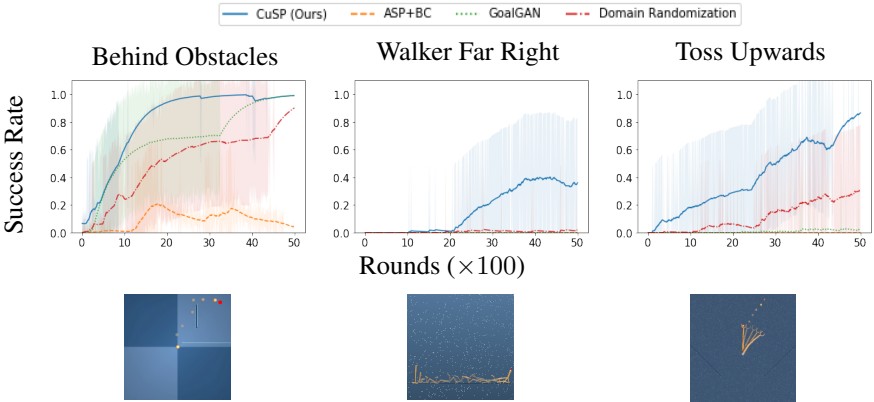

Figure 6: Success rates for three more challenging goals in Point Mass Obstacle, Walker, and Manipulator, and corresponding sample trajectories. In Point Mass we learn to navigate around walls faster than other methods, in Walker we learn to run quickly (albeit by using its torso) whereas other methods fail to succeed at all, and in Toss we learn to aim upwards accurately.

that the domain randomization baseline is particularly well suited for this OOD random evaluation. The baseline is trained on good coverage of the in-distribution goal space, which is beneficial for generalization to random goals in the navigation and manipulation tasks, but is less helpful for the more asymmetric goal spaces like Walker and Toss. For ablations on ASP, see Appendix B.5.2. We also ablate over each contribution of our method as detailed in Table 1 in Appendix B.5.1. Specifically, we ablate the regret replay buffer, the dynamic regret updates, entropy regularization, and symmetrization. We find that only using the regret objective from PAIRED (Dennis et al., 2020) is insufficient, even with naively applying symmetrization, and that both entropy regularization and an updated replay buffer are crucial for high evaluation task success.

## 4.2 TASK-SPECIFIC SKILLS

Next, we examine if CuSP can induce emergent skills in learners for tackling harder, environment specific goals. Below, we enlist 3 representative examples in the more asymmetric environments where certain goals are clearly more challenging to solve. As seen in Fig. 6, our method is able to improve upon success rates on these goals.

- Point Mass Obstacle – *Behind Obstacle*s: requires learning to move around the walls to the edge of the training space. Here, we learn to succeed much faster than the other methods.
- Walker – *Far Right*: requires learning to move forward quickly to reach a goal far out of the training space. Here, we learn to succeed at the task where all other baselines fail, suggesting that our method produces a useful curriculum for learning challenging locomotion tasks while the others cannot, or require many more samples to do so.
- Toss – *Upwards*: requires learning to toss the ball accurately upwards outside of the training space. Here, we learn to succeed much faster than the other methods.

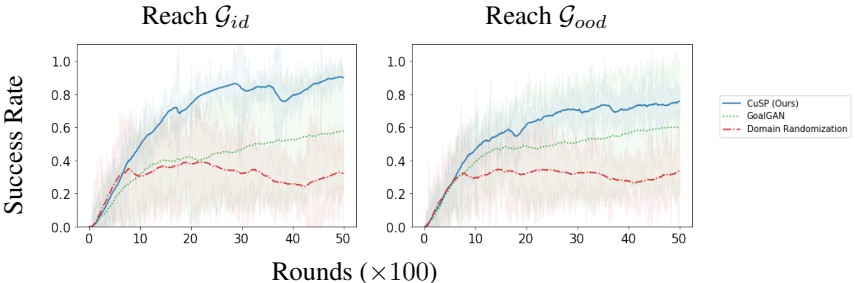

Figure 7: Success rates for Reach with a third impossible dimension added to the specified goal space. We test on both in distribution and out of distribution randomly sampled goals in the 3D goal space, where the third dimension is not feasible in the environment. Generally we find CuSP is least affected while the baselines generally performs worse with the higher dimensional space.

### 4.3 MISSPECIFIED GOAL DIMENSIONS

Lastly, we investigate if the goal generator methods (CuSP, GoalGAN, and DR) can handle the case where parts of the specified goal space is misspecified. To do this, we append an extra dimension to the goal space, bounded between $[-1, 1]$, and evaluate performance on $\mathcal{G}_{id}$ and $\mathcal{G}_{ood}$ with randomly sampling the third dimension during evaluation. We use Reach as a case study, and as seen in Fig. 7, agents trained using CuSP's goal curricula are less affected by the increased goal space while the baselines plateau at a lower success rate. In particular, the DR baseline is most adversely affected by the higher dimensionality. This suggests that while random sampling provides good coverage and is amenable to more simple goal spaces, as seen above, the induced curricula degrades in quality with increasingly complex goal spaces. Note that we exclude ASP from these experiments as it does not use a pre-specified goal space and only sets achieved states as goals.

## 5 RELATED WORK

We summarize four main thrusts of work within automatic curriculum generation.

**Modeling Goal Distributions.** Many previous works have tackled the problem of generating "appropriately" difficult goals for an autocurriculum. In discrete environments, Campero et al. (2021) propose an adversarial goal-generator that is rewarded when the learning agent achieves the goal with a thresholded level of effort. However, they rely on a heuristic time limit hyperparameter to determine 'suitable effort' (i.e. goal feasibility) and hyperparameters for the teacher reward, whereas our approach directly uses regrets from multiagent interactions to automatically gauge feasibility and provide a reward signal to the teacher. In continuous domains, Zhang et al. (2020) samples goals that maximize the uncertainty of the learner's Q-function, and Nair et al. (2018) learn a latent representation to sample 'imagined' goals from. GoalGAN (Florensa et al., 2018) trains a GAN to generate goals of intermediate difficulty for the agent, using a discriminator to predict whether the generated goals would be appropriate by thresholding success rates. These methods use a heuristic measure of challenging goals, while we leverage the multi-agent setting to automatically find interesting goals. Similarly to GoalGAN, the Setter-Solver paradigm (Racaniere et al., 2020) trains a goal setter whose objectives are a balance of goal validity, feasibility, and coverage, as well as a discriminator for predicting goal feasibility. In contrast, our approach does not need to explicitly weigh the contributions of each loss or incorporate discriminators.

**Multi-Agent Curriculum Learning.** The idea that interactions between multiple agents can lead to emergent curricula has been proposed in prior works (Leibo et al., 2019; Baker et al., 2019) and relates to the theory of mind (Rabinowitz et al., 2018; Grover et al., 2018a;b) postulating human learning via social interactions. One parametrization of such interactions is through designating another agent(s) as a teacher who aims to select an appropriate curriculum for the student. For task-based curricula, Matiisen et al. (2019) train a teacher policy for selecting sub-tasks for the student, and Graves et al. (2017); Portelas et al. (2019) treat task selection as a bandit problem for the teacher.

Rather than having an explicit teacher, Asymmetric Self-Play (ASP) (Sukhbaatar et al., 2018b) automatically generates a curriculum for exploration by training a similarly parametrized agent, Alice, to

demonstrate tasks that the learning agent, Bob, struggles to complete. OpenAI et al. (2021) extend ASP to robotic manipulation tasks by incorporating an imitation learning loss for Bob to imitate Alice on challenging goals. As Alice is no longer setting the goals, we do not make use of behaviour cloning in CuSP. While they demonstrate zero-shot generalization to a diverse set of target tasks, relying on Alice to set the goals can also be incredibly sample inefficient, as we saw in our evaluations. Sodhani & Pahuja (2018) propose a memory-augmented version of ASP by providing the learners an external memory to speed up exploration, which is analogous to our method training with off-policy learners. Sukhbaatar et al. (2018a) extends ASP to a hierarchical learning framework by learning subgoal representations. Competitive Experience Replay (Liu et al., 2018) generates an exploratory competitive curriculum through penalizing Alice if visiting states Bob has visited, but rewards Bob for visiting states found by Alice.

**Environment Design.** Another method for inducing curricula is through generating a sequence of environments to learn in. Procedural level generation is a common approach for developing games, and can lead to increased generalizability (Justesen et al., 2018). POET (Wang et al., 2019) uses a population of adversaries to generate increasingly challenging environments for a population of learning agents. PAIRED (Dennis et al., 2020) proposes optimizing an adversarial environment designer through minimax regret between two learning agents, inspiring the objective we use for our goal generators. However, this approach does not account for catastrophic forgetting and can suffer from sample inefficiencies from using an on-policy method and in cases where the regret landscape is flat. That said, environment design approaches can be used in a complementary way with our approach, extending it beyond goal generation.

**Skill Discovery.** A closely related method for curriculum generation is through unsupervised skill discovery, where an agent explores an environment to automatically discover skills that can be composed to achieve a variety of tasks. Prior methods have used information theoretic objectives to encourage skill acquisition that covers a diverse set of behaviours (Eysenbach et al., 2018) or generate a multi-task distribution (Gupta et al., 2020). For robotics tasks, (Hausman et al., 2018) propose a method for learning a latent skill space that can be composed and is transferable between tasks, DADS (Sharma et al., 2019) proposes a combination of model-based and model-free RL that optimizes for discovering predictable skills that can be composed by a planner, and Play-LMP (Lynch et al., 2020) proposes using self-supervision across play data for skills discovery. Another method for curriculum generation is through using hindsight (Andrychowicz et al., 2017; Li et al., 2020). These works are complementary to ours as our method aims to induce a curriculum through sparse goal generation, which can be augmented with unsupervised skill discovery.

# 6    CONCLUSION

We propose a new method, CuSP, for automatic curriculum generation based on symmetric extension of self play approaches for non zero-sum games. Our proposed method aims to address weaknesses of prior approaches by being designed to specifically consider a balance of progressive exploration and anti-catastrophic exploitation. To this end, we make use of two off-policy students and two off-policy regret-maximizing goal generators, each of which is more 'friendly' to one of the students. As a result, we generate goals that are a) challenging yet feasible, b) diverse, and c) revisited to mitigate anti-forgetting. We qualitatively highlight these desired qualities in goal generation compared to prior methods, demonstrate quantitative gains at generalizing to out-of-distribution unseen goals, and find some emergent task-specific skills.

**Limitations and Future Work.** While our method shows promise for addressing key components of generating useful curriculum, there is still significant room for improvement. One limitation of our current approach is that it depends on specifying a goal space for the generator to act in, which was not necessary in the original ASP method. That said, specifying a goal space also allows us to imbue the system with human priors for what goals should or should not be valid, whereas ASP may need post-hoc goal filtering if Alice ends up in an undesirable state.

To further tackle automatic curriculum generation, future work should explore scenarios where goal spaces are not fully pre-specified, as well as extending to high dimensional goal spaces (e.g. images). Furthermore, another interesting avenue would be extrapolating CuSP to a larger number of student and teacher agents, which can potentially provide benefits through better goal space coverage and giving a more accurate estimate of feasibility.

ACKNOWLEDGEMENTS

Thanks to Eugene Vinitsky for helpful discussions, and Hao Liu and Olivia Watkins for feedback on drafts. This work was supported by the Center for Human-Compatible Artificial Intelligence and a FAIR-BAIR collaboration between UC Berkeley and Meta.

ETHICS STATEMENT

Automatic curriculum generation has potential as a method for learning complex agent behaviours in an unsupervised way. Requiring less direct supervision (eg. through hand-tuned, shaped reward functions) can lead to more robust AI systems for real-world applications. At the same time, it is important to consider potential impacts unsupervised reinforcement learning, as it may lead to unintended behaviours. By incorporating human engineering and priors through goal space specification in the goal generative methods, like CuSP, we can help restrict the set of goal spaces to more desirable (e.g. safe) behaviours. While our empirical studies have been done in simulated systems, careful and thorough analysis of learned behaviours should be done before deploying CuSP on real world systems.

REPRODUCIBILITY STATEMENT

Code is available at `https://github.com/yuqingd/cusp`. For experimental environment details, see Appendices B.1, B.2. For hyperparameters, see Appendix B.3. For ablations of CuSP and ASP, see Appendices B.5.1 and B.5.2 respectively.

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

## A    DESIRED GOAL GENERATION QUALITIES

We begin by qualitatively examining the key properties of the curriculum of goals generated in Point Mass Obstacle. We see that qualitatively our method (Fig. 8a) proposes goals with more progressive exploration than either ASP (Fig. 8d) (which seems to get stuck on the walls, limited by Alice's ability to explore) or GoalGAN (Fig. 8c) (which seems to get stuck more along the borders of the space, only prioritizing difficult goals). The DR generator (Fig. 8b) has the opposite result, having great coverage but without any consideration for feasibility or difficulty. Our method manages to balance the exploration with progressive feasibility, where we see CuSP proposing goals more towards the corners at first, then progressively moving towards the center and covering the space more thoroughly, with slightly more density in the top right where the maze is. We hypothesize there is more concentration in the corners initially when the goal generators are untrained and proposing goals towards the edges of the goal constraints.

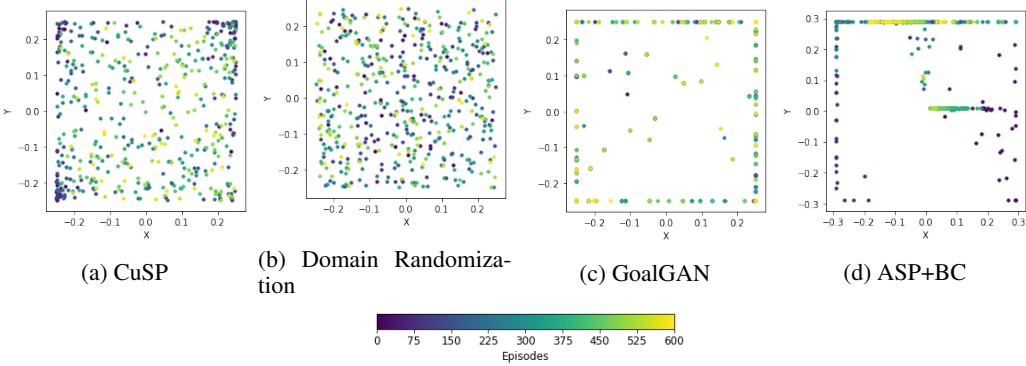

(a) CuSP     (b) Domain Randomization     (c) GoalGAN     (d) ASP+BC

Figure 8: Progressive goal generation plots for Point Mass Obstacle after 500 rounds across all four methods. We see more progressive diversity from CuSP, whereas GoalGAN mostly stays on the outer border and ASP+BC gets stuck on the environment walls, limited by Alice's abilities.

## B    EXPERIMENTS

**Compute.** To train each method in each environment, we use 1 NVIDIA V100 GPU per seed in an internal cluster. This amounts to an approximate total of 400 GPUs for the final evaluations and hyperparameter sweeps. We additionally did extensive prototyping on the Point Mass environment before scaling to the more complex environments, but do not have a reasonable estimate for how much compute was expended in that phase of the project.

### B.1    TOY EXPERIMENTS

For the toy evaluations in Figures 2 and 3, we evaluate on the landscape

$$\text{regret}(x, y) = \begin{cases} -((x+.1)^2 + (y-.1)^2) & \text{if} (x+.1)^2 + (y-.1)^2 < .01 \\ -.01 & \text{otherwise} \end{cases}$$

For Adam, we use the default configuration from PyTorch (lr=0.001, $\beta$s = (0.9, 0.999), $\epsilon$=1e-8, no weight decay). For PPO, we use the default configuration from (Kostrikov, 2018). We found our trends to hold robustly for a fairly broad choice of lr's and both with/without GAE (Schulman et al., 2018). In the toy experiments, we carry out 5000 steps of goal proposal with one gradient update for every method at each step. For the non-stationary experiment in Figure 3, we diagonally perturb the center of the regret landscape from top left quadrant (initialized at (-0.1, 0.1)) to the bottom right quadrant at a constant rate of 2e-4 every round. Hence, at the end of 2500 rounds, the center of the landscape is at the origin and at the end of 5000 rounds, the center of the landscape is at (0.1, -0.1).

## B.2 ENVIRONMENT SPECIFICS

Table 2 describes the reward configurations we used in our environments. For the manipulation environments, we also incorporated HER (Andrychowicz et al., 2017). $\epsilon$ is the tolerance for success (i.e. a goal $g$ is successfully achieved at state $s$ if the normed distance $L_2(s - g) < \epsilon$). We terminate the episode early if successful.

| Env | Reward | Use HER (Andrychowicz et al., 2017) | $\epsilon$ |
|---|---|---|---|
| Point Mass Maze | $-L_2$ | | .05 |
| Walker | $-L_2$ | | .1 |
| Reach | $-L_2$ | ✓ | .1 |
| Toss | $-L_2$ | ✓ | .1 |
| Pick and Reach | 0 if at goal, -1 otherwise | ✓ | .1 |

Table 2: Environment Reward Configurations

We modify the default Point Mass environment by adding in walls to the top right corner and initialize the agent in between the walls 10% of the time to help with exploration. We modify the manipulator environment by always initializing with the ball in the gripper and with the gripper in an upright position. We use an episode length of 1000 steps for Point Mass Obstacle and Walker and an episode length of 100 steps for Reach, Toss, and Pick.

Specific definitions of in-distribution and out-of-distribution goal spaces for each environment are in Table 3.

| Env | $\mathcal{G}_{id}$ | $\mathcal{G}_{ood}$ |
|---|---|---|
| Point Mass Maze | $[-.25, .25] \times [-.25, .25]$ | $[-.3, .3] \times [-.3, .3]$ |
| Walker | $[-1, 10] \times [-.2, .2]$ | $[10, 20] \times [-.2, .2]$ |
| Reach | $[-.25, .25] \times [.2, .6]$ | $[-.4, .4] \times [.1, .8]$ |
| Toss | $[-.5, .5] \times [.5, 1.3]$ | $[-.6, .6] \times [.5, 1.4]$ |
| Pick and Reach | $[.5, .9] \times [.5, .7]$ | $[.4, 1] \times [.5, .8]$ |

Table 3: Goal spaces for each environment.

## B.3 HYPERPARAMETERS

We train the learners and goal generators using the default SAC configuration and implementation from (Yarats & Kostrikov, 2020), as listed in Table 4, with separate networks for the actor and critic.

| Parameter | Value |
|---|---|
| $\gamma$ | .99 |
| Initial $\alpha$ | .1 |
| $\alpha$ LR | 1e-4 |
| Actor LR | 1e-4 |
| Critic LR | 1e-4 |
| Batch size | 1024 |
| Critic hidden dim | 1024 |
| Critic hidden depth | 2 |
| Actor hidden dim | 1024 |
| Actor hidden depth | 2 |

Table 4: SAC Hyperparameters

To scale the output of the goal generators to the correct action space, we take the Tanh scaled output and rescale each dimension to the goal space. We use Appendix C of (Haarnoja et al., 2018) to modify the log loss for SAC accordingly. To train the goal generator, we update after each round (i.e. per goal proposed) with 100 gradient updates. As the learners are updated once per time-step in the environment (100-1000 updates depending on the environment), we increased the update frequency for the goal-generator to keep pace with the learner updates since the goal generator operates on a single time step per episode.

### B.4 BASELINES

We reimplement each baseline by comparing against any available public code and matching each paper's recommended training hyperparameters as closely as possible within our compute limitations.

**Off vs. On Policy Learners** In ASP+BC(OpenAI et al., 2021), Alice and Bob are trained using PPO. We also implemented this as a baseline but found that the agents were unable to reach high success at the same rate as our SAC-trained agents. For a comparison that's more favorable to the baselines, we report the success rates from training with the same agent configuration (ie. Bob trained with SAC) in the main paper to isolate learning differences to the goal generation method.

### B.5 ABLATIONS

#### B.5.1 CUSP ABLATIONS

**SAC Motivation** Without symmetrization, the regret-based goal generation objective is the same as the proposed objective in PAIRED Dennis et al. (2020). However, our contribution extends beyond the symmetrized objective as our proposed method reframes the goal generation process into a entropy-regularized agent with a memory buffer – which motivated designing our method around SAC. PAIRED uses PPO to optimize their environment generators, which we found was not sample efficient enough for our tasks (i.e. success rates were too low), as our toy experiment in Figure 2 suggested.

Here we highlight a case study with the Toss task into why the entropy regularization and replay buffer components are important. Building up from just the regret objective as in PAIRED, we have in Figure 9:

1. $\beta = 1, \alpha = 0$, no replay buffer – Regret objective only for a single goal generator, as in PAIRED.

2. $\beta = 1, \alpha = 0$, no replay buffer, symmetrized – Regret objective with two symmetrized goal generators. Symmetrization seems to improve performance slightly, but is not sufficient alone.

3. $\beta = 1, \alpha = 0$ – Regret objective with a replay buffer for the goal generator. Also improves performance slightly, although not as much as symmetrization.

4. $\beta = 1$ – Regret objective with a replay buffer and entropy regularization. This is particularly helpful for the random OOD evaluation as the greater goal diversity improves multi-goal performance on the multi-goal evaluation task, as expected.

5. $\beta = 1$ symmetrized – Regret objective with a replay buffer and entropy regularization, and symmetrized goal generators.

6. $T = 300, \beta = .1$, symmetrized (Full CuSP) – Regret objective with a dynamically updated replay buffer and entropy regularization, and symmetrized goal generators.

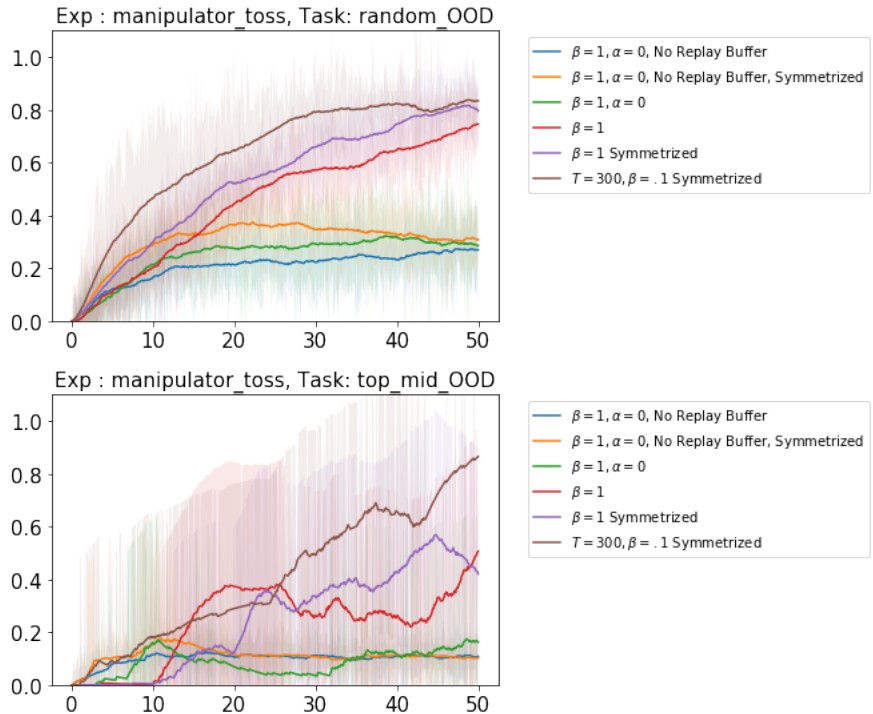

Figure 9: Full ablations of all components of CuSP from Table 1. Success rates are averaged across 3 seeds.

Looking at a sample of the generated goals in Figure 10, we find a stark qualitative improvement in goal diversity between the regret-only PAIRED condition and CuSP. In particular, in the baseline the generated goals converge only on the corners.

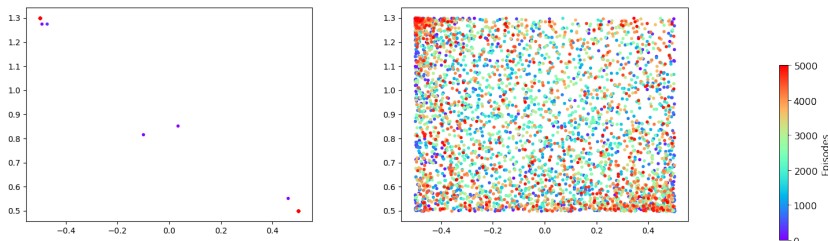

Figure 10: Goal plots after 2000 rounds for Toss. The left plot shows the goals generated in the $\beta = 1, \alpha = 0$, No Replay Buffer condition and the right plot shows the goals generated in the $T = 300, \beta = .1$, Symmetrized condition from Figure 9. Red indicates more recent goals.

**Regret update and Symmetrization Ablations** As we use the learners' critics for estimating the current regret for a particular goal, we need to be careful about how we use the critic data since it can be inaccurate, especially towards the beginning of training. We introduce two hyperparameters: one for specifying at which episode we begin regret updates $T$, and a weighting parameter $\beta$ for how much to update the regrets by. We use the update rule

$$\text{regret} = \beta \cdot \text{old regret} + (1 - \beta) \cdot \text{new regret}$$

We sweep across $T = 50, 100, 200$ and $\beta = .1, .5, .9$ to select the best performing hyperparameters based on average performance on the training goal set, $\mathcal{G}_{id}$. In Figure 11, we ablate over the different components of our method: CuSP+SAC with no entropy regularization ($\beta = 1, \alpha = 0$),

CuSP+SAC only ($\beta = 1$), CuSP+SAC+Symmetrization ($\beta = 1$), and CuSP+SAC+Regret Updates+Symmetrization to investigate the effects of each component. In most of the tasks, aside from Reach on random out of distribution goals, symmetrization improves performance. In particular, we see larger gains on tasks where $\beta = 1$ performance is lower to begin with, such as Walker. Further gains from stale regret updates vary from being very small on tasks where $\beta = 1$ performance is already high (eg. for Point Mass Obstacles or Toss on random OOD goals), to larger improvements on the harder skill tasks (eg. Behind Obstacles or Walker Far Right goals). Interestingly, we also note that while entropy regularization is crucial for most of the environments, setting $\alpha = 0$ seems to be helpful for Walker.

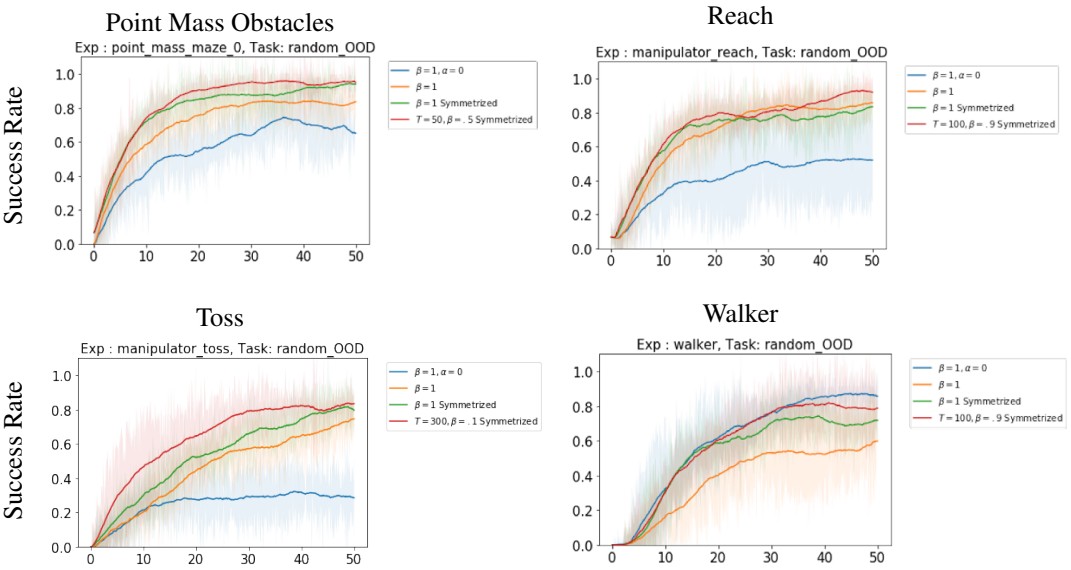

Figure 11: Success rates across environments on random goals sampled from $\mathcal{G}_{ood}$, ablating for SAC-based goal generators, symmetrization and stale regret updates. We generally find that having both components improve success rates across the tasks, although the gains vary depending on how challenging the environment is to begin with. In particular, having both the regret updates and symmetrization helps most on the more challenging Walker tasks and the skill-specific tasks.

**Multiagent Motivation.** Here we investigate whether it is sufficient to formulate the CuSP game with only a single learner. Rather than using regret defined as the difference in returns of two agents, we look at the difference in the returns of a single agent across two rollouts to identify if stochasticity in the policy alone is sufficient for providing the goal generator signal. As a case study we look at the Toss task in Figure 12, where we find the success rate increases much more slowly with a single agent. We hypothesize that a challenge with only using a single agent fro defining regret is that the regret landscape is much more flat – i.e. the regret is more likely to be very small and not provide much signal to the goal generators, as the stochasticity of a single agent policy is comparatively lesser than two separate agents. As a result, the goals are less likely to be a useful curriculum for the agent.

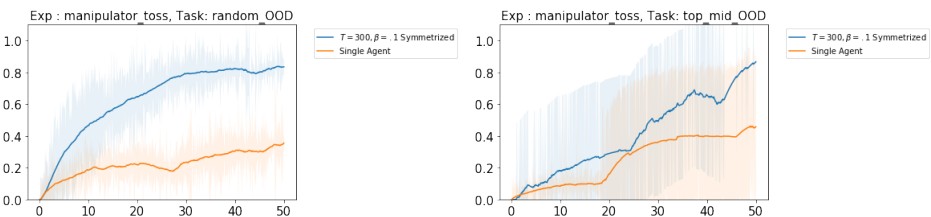

Figure 12: Comparison of CuSP with separately initialized and updated Alice and Bob, or a single Alice. Success rates are averaged across 3 seeds.

### B.5.2 ASP ABLATIONS

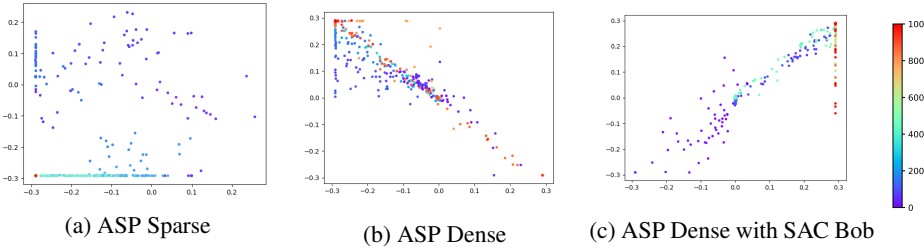

(a) ASP Sparse      (b) ASP Dense      (c) ASP Dense with SAC Bob

Figure 13: Progressive goal generation plots for the Point Mass environment after the first 1000 episodes, where each goal is the final position of Alice at the end of the episode and both agents are initialized at the center of the domain. We see that with the sparse version of ASP, the proposed goals converge on a single corner over time and gets stuck, whereas the dense implementation leads to more diverse goals.

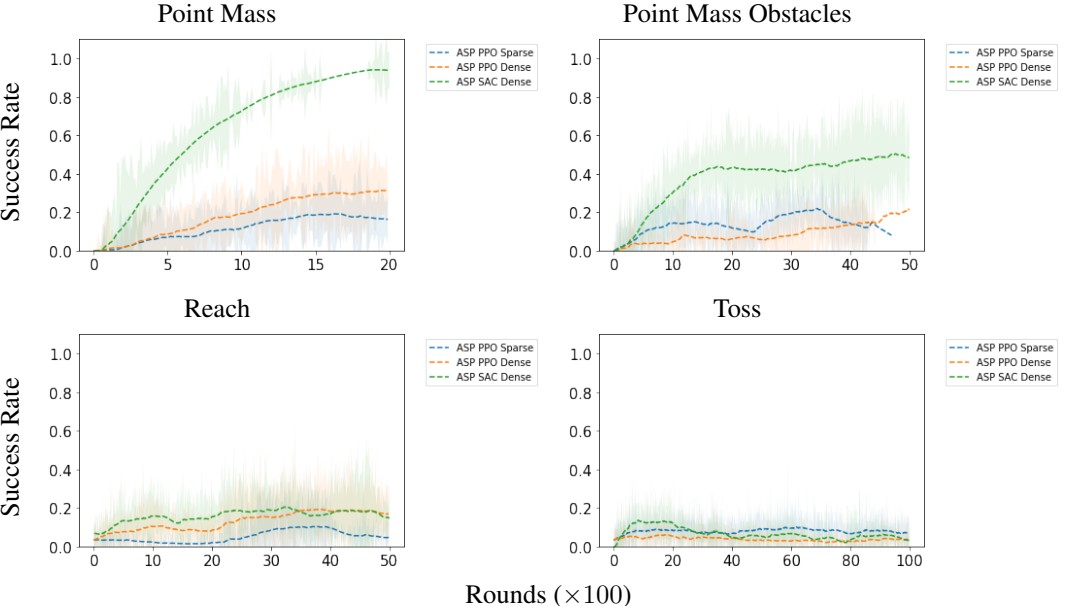

Figure 14: Success rates across environments where ASP achieved $> 0$ success rates on random goals sampled from $\mathcal{G}_{ood}$ across each of the ASP variants. For final results in the paper, we report ASP+BC (Dense, SAC).

In the original implementation, ASP+BC uses PPO to optimize both Alice and Bob. We find this very suboptimal for the environment setups in this work. For empirical validation, here we report results using 3 variations of ASP+BC:

- ASP+BC (Sparse): Alice is rewarded +1 at the end of an episode of Bob does not succeed, or receives 0 reward otherwise. Both Alice and Bob are optimized using PPO.

- ASP+BC (Dense): Alice is rewarded negative Bob's reward at each transition. Both Alice and Bob are optimized using PPO.

- ASP+BC (Dense, SAC): Alice is rewarded negative Bob's reward at each transition. Alice is optimized with PPO while Bob is optimized with SAC.

For all cases, we use the behavioural cloning (BC) mechanism proposed in Section 3.2 of (OpenAI et al., 2021), where Bob is updated with the same clipped behavioural cloning loss on trajectories where it does not successfully achieve Alice's proposed goal.

The results are shown in Figure 14. We observe that with a sparse reward, Alice would tend to get stuck with proposing similar goals in a particular location. While Bob learns to succeed at that single goal, the lack of diversity causes Bob to perform poorly at generalization to novel goals in $\mathcal{G}_{ood}$, as seen in the evaluations. To overcome this, we incorporated a dense reward implementation and swapped to optimizing for Bob's policy using SAC. Our results presented in Section 4 use this stronger baseline for comparison against CuSP.

## C  THEORETICAL ANALYSIS

Our overall framework contains 4 players: two goal-conditioned agents, Alice $\pi_A$ and Bob $\pi_B$, and their corresponding goal generators, $G_A$ and $G_B$. Further, let $\Pi$ and $\mathcal{G}$ denote the space of policies and goal generative models respectively. At any given round, let $g_A \sim G_A$ and $g_B \sim G_B$ denote the corresponding goals sampled from $G_A$ and $G_B$ respectively. Accordingly, we can define two regrets for the sampled goals:

$$\mathfrak{R}^{G_A}(g_A, \pi_A, \pi_B) = R(g_A, \pi_A) - R(g_A, \pi_B) \tag{1}$$

$$\mathfrak{R}^{G_B}(g_B, \pi_B, \pi_A) = R(g_B, \pi_B) - R(g_B, \pi_A) \tag{2}$$

where $R(g, \pi_A)$ and $R(g, \pi_B)$ denote the empirical discounted sum of rewards for Alice and Bob respectively on a goal $g$. Now, we define the objective function $f$ below:

$$f(g_A, g_B, \pi_A, \pi_B) := \mathfrak{R}^{G_A}(g_A, \pi_A, \pi_B) - \mathfrak{R}^{G_B}(g_B, \pi_B, \pi_A). \tag{3}$$

The overall objective function for CuSP can be written as:

$$\min_{G_B \in \mathcal{G}, \pi_B \in \Pi} \max_{G_A \in \mathcal{G}, \pi_A \in \Pi} \mathbb{E}_{g_A \sim G_A, g_B \sim G_B}[f(g_A, g_B, \pi_A, \pi_B)] := F(G_A, G_B, \pi_A, \pi_B). \tag{4}$$

To analyze the above objective, we define *coordinating goal generator-solver teams* $(G, \pi)$ as a pair of goal generator $G \in \mathcal{G}$ and goal conditioned agent $\pi \in \Pi$. Accordingly, we can interpret CuSP as a 2 player zero-sum game between two coordinating goal generator-solver teams $(G_A, \pi_A)$ and $(G_B, \pi_B)$ defined via goal-conditioned agents $\pi_A \in \Pi$ (Alice) and $\pi_B \in \Pi$ (Bob) along with their corresponding friendly goal generators $G_A \in \mathcal{G}$ and $G_B \in \mathcal{G}$ respectively.

**Definition C.1.** *(CuSP game) The CuSP game is a 2 player zero-sum game between two coordinating goal generator-solver teams $(G_A, \pi_A)$ and $(G_B, \pi_B)$ defined via:*

- *Alice $\pi_A \in \Pi$ and its friendly goal generator $G_A \in \mathcal{G}$ and*

- *Bob $\pi_B \in \Pi$ and its friendly goal generator $G_B \in \mathcal{G}$.*

*The game is zero-sum with payoff function given as $F(G_A, G_B, \pi_A, \pi_B)$ for $(G_A, \pi_A)$, and as $-F(G_A, G_B, \pi_A, \pi_B)$ for $(G_B, \pi_B)$.*

Next, we state conditions under which a Nash Equilibrium exists for coordinating goal generator-solver agents in CuSP. The conditions depend highly on whether the game strategies (given by the product of goal generator space and policy space) are finite or continuous.

**Proposition C.2.** *(Finite Games) Let $(G_A, \pi_A) \in \mathcal{G} \times \Pi$ and $(G_B, \pi_B) \in \mathcal{G} \times \Pi$ be two coordinating goal generator-solver agents for CuSP defined over finite goal space $\mathcal{G}$ and policy space $\Pi$. Then, there exists a mixed strategy Nash Equilibrium for the CuSP game.*

*Proof.* Since $\mathcal{G}$ and $\Pi$ are finite, we know that the product space $\mathcal{G} \times \Pi$ is also finite. The result then follows directly from Nash's Theorem (Nash, 1951). $\quad\square$

**Proposition C.3.** *(Continuous Games) Let $(G_A, \pi_A) \in \mathcal{G} \times \Pi$ and $(G_B, \pi_B) \in \mathcal{G} \times \Pi$ be two co-ordinating goal generator-solver agents for CuSP defined over continuous goal space $\mathcal{G}$ and policy space $\Pi$. Further, let $\mathcal{G}$ and $\Pi$ be nonempty compact metric spaces and the payoff function $F$ (Eq. 4) be continuous. Then, there exists a mixed strategy Nash Equilibrium for the CuSP game.*

*Proof.* Since $\mathcal{G}$ and $\Pi$ are compact, we know by Tychonoff's theorem (Tychonoff, 1930; Willard, 2012) that the product space $\mathcal{G} \times \Pi$ is also compact. Combined with the fact that $F$ is assumed to be continuous, we can apply Glicksberg's Theorem (Glicksberg, 1952) to finish the proof. $\qquad\square$

Since we have established conditions for the existence of Nash equilibria for the CuSP game, we know from the minimax theorem that we can recover the Nash equilibirum solution by optimizing Eq. 4.

**Note 1:** In practice, our algorithms for optimizing Eq. 4 might not satisfy the necessary assumptions for the theoretical results. For example, we optimize the parameters of generators and policies specified as neural networks as opposed to optimizing in the space of functions directly. Further, we consider a single-sample Monte Carlo estimate for the CuSP objective in Eq. 4 and optimize the goal generators and policy networks sequentially for a fixed number of gradient updates in every round using RL algorithms such as SAC. While these protocols are standard practice even in related works, deriving theoretical guarantees in such scenarios is extremely challenging and an active area of theoretical research with many open questions.

**Note 2:** If $\pi_A = \pi_B$, then $r_A(g) = r_B(g)$ for all $g$, so all corresponding regrets will also be zero and thus give no signal to the goal generators. The key point to note here is that in practice, the goal generators have an exploration component that can avoid getting stuck, e.g., this can be achieved via $\epsilon$-greedy or via SAC which has an exploration bonus (as done in CuSP).

Concretely, let $\pi_A = \pi_B = \pi_{rand}$ be randomly initialized agents. We claim that at initialization, $(G_A, \pi_{rand})$, $(G_B, \pi_{rand})$ will not be in a Nash Equilibrium except in the case where a random policy is already able to achieve all possible goals. To see why, suppose for the sake of contradiction that the agent teams $(G_A, \pi_{rand})$, $(G_B, \pi_{rand})$ are at a Nash Equilibrium. That is, the teams have the joint best responses to each other. Let $\mathbb{G}$ be the set of all possible goals and $\mathbb{G}' \subseteq \mathbb{G}$ be the set of goals that $\pi_{rand}$ can solve.

If $\mathbb{G}' = \mathbb{G}$, then $\pi_A, \pi_B$ are already maximally competent and are therefore done learning. Note that this is also what we hope to converge to eventually during training, and the two policies matching can correspond to a Nash Equilibrium in such a case. At this point $\mathfrak{R}(g_A, \pi_A, \pi_B) = \mathfrak{R}(g_B, \pi_B, \pi_A) = 0$ for all $g_A, g_B$ so neither $G_A, G_B$ can increase their payoffs, and since $\pi_A, \pi_B$ are parameterized identically, if either policy is able to increase its payoff on some $g$ we can identically update the other policy such that $\pi_A = \pi_B$ until neither policy is able to improve its payoff further either.

If $\mathbb{G}' \subset \mathbb{G}$, then for either team we can pick any single goal $g^* \in \mathbb{G} \setminus \mathbb{G}'$ and learn a policy $\pi^*$ capable of achieving $\{g^*\} \cup \mathbb{G}'$ using policy iteration (e.g., with greedy policy improvement assuming no forgetting of past goals). Such a $g^*$ will eventually be proposed as long as the goal generators have an exploratory component. This means there is a strategy $\pi^*$ that acquires equal utility to $\pi_{rand}$ on $\mathbb{G}'$ and higher utility on $g^*$. Thus there exists a strictly better strategy for either team $(G_A, \pi_{rand})$, $(G_B, \pi_{rand})$ by switching from $\pi_{rand}$ to $\pi^*$, which contradicts that the players are already at a Nash Equilibrium to begin with.

