# OpenReview forum: "It Takes Four to Tango: Multiagent Self Play for Automatic Curriculum Generation"
_ICLR.cc/2022/Conference — ICLR 2022 Poster_

### Official Review · Reviewer_A4MW · 2021-11-01

**Correctness:** 3
**Technical Novelty And Significance:** 3
**Empirical Novelty And Significance:** 3
**Recommendation:** 8
**Confidence:** 4

**Main Review:**

Main strengths

* I found interesting the idea of multiple "teachers" and agents to account for catastrophic forgetting while exploring for new goals.
* I really liked the idea of updating the regrets from the replay buffer using the new evaluations from the critic, as far as I know this is new and quite nice
* Authors include quick and nice illustrations in Figures 1 and 2 illustrating the advantages of SAC and using updated regrets in the replay buffer
* Multiple benchmarks used

Main weaknesses

My bigger concerns with this work come from not well detailed parts/assumptions, the narrow comparison with AMIGO, the fact of highlighting the benefits of using a symmetric method with 4 agents but the lack of analysis of using larger numbers (e.g., 6 or 8) and leaving that part for future work. I detail all of these below:

---
* I didn't find a clear explanation of what is the action space of the goal-generation agents, authors comment that they need to be predefined and embeded with human priors of which goals are acceptable. This is an important limitation for which I found little discussion, an explanation of how these are defined in the experiments would be very beneficial.

* How are critic and actor implemented, separated? do they share the same networks? since here the critic plays an important role updating the replay buffer it would be beneficial to have a closer look on this

* In section 2 is mentioned that the goal distribution is separated into train and test but it is never clearly stated how this is done

* In page 4, first paragraph you mention that for the goal generators you add a latent noise variable to the state, why?

* In your method there is not really a "main solver" or agent, when reporting your results which one you pick? why?

---
* You cite the recent work from Campero et al. 2021 about AMIGO which also uses a goal generator agent and a goal achiever agent, it could be interpreted as a 2-agent version of what is presented here but in discrete spaces, I missed then a more in depth comparison with their work in benchmarks or , at least, theoretically.

---
* Since your work is not the first one exploring the idea of an agent looking for achievable yet challenging goals using a goal-generator agent and a reacher, the main contribution seems to be in the introduction of multiple of these and how they help each other, but then why stop in four? From the very beginning your work raises that question, thus is kind of disappointing that it is just left for future work.


---

### Post-discussion and updated version

I believe that authors have done a good work in the discussion and answering my concerns. I would have liked to see an analysis of the effect and equilibrium with systems of larger number of agents, but the current work has more than sufficient analysis and contributions on its own as reflected in the additional ablations and discussion incorporated in the new version. I have updated my scores accordingly



**Summary Of The Paper:**

This paper introduces and evaluates a new unsupervised reinforcement learning method for curriculum learning. It aims to training agents in continuous environments with increasingly difficult tasks with a 4-agent algorithm with two goal-agents or teachers that have their respective "favourite" student that they want to excel over the other teacher's favourite.

**Summary Of The Review:**

The work introduces nice concepts and ideas that are not completely well supported nor fully explored. Thus although I like the  idea I am reluctant to give an accept at its current state

### Post-discussion and updated version
I think that the updated version supports much better author's statements and reflects better their contributions

---

> ### Author Response · Authors · 2021-11-16
> **Response to A4MW (1/2)**
>
> We thank the reviewer for their detailed comments and am glad they found the idea interesting.
>
> We address specific comments below:
>
> > I didn't find a clear explanation of what is the action space of the goal-generation agents, authors comment that they need to be predefined and embeded with human priors of which goals are acceptable. This is an important limitation for which I found little discussion, an explanation of how these are defined in the experiments would be very beneficial.
>
> The action space of the goal generator consists of the set of possible goals for a particular environment. For example, in a pick and place task, the goal space can consist of a variety of goal object positions on a table, but not object positions on the ground, since pushing objects off a table is undesirable behaviour. In ASP (OpenAI et al.), the authors propose filtering generated goals post-hoc based on whether they were ‘valid’ or not (e.g. off the table). Rather than filtering after the rollouts, which can waste samples, we instead predefine a valid goal generation space as in prior goal generation works (e.g. GoalGAN). For our particular set of environments, we define goal spaces that consist of feasible states for each environment (e.g. we do not define the point mass goal space to exceed the boundaries of the space, or the goal space for the reaching task to be beyond the reachable bounds). **We have updated Appendix B.2 with Table 3 showing the specific goals spaces for each environment.**
>
> > How are critic and actor implemented, separated? do they share the same networks? since here the critic plays an important role updating the replay buffer it would be beneficial to have a closer look on this
>
> We directly use the implementation from https://github.com/denisyarats/pytorch_sac, where the actor and critic are separated.
>
> > In section 2 is mentioned that the goal distribution is separated into train and test but it is never clearly stated how this is done
>
> We show illustrations of the train and test goal spaces in Figure 4 and have **updated Appendix B.2 with Table 3 showing the separation between ID and OOD goal spaces.** In general, the test goal space is a superset of the training one, including goal states that are harder to reach / further out than the training set.
>
> > In page 4, first paragraph you mention that for the goal generators you add a latent noise variable to the state, why?
>
> This design choice builds on standard practice in generative models (e.g. GANs, VAEs) where a latent noise variable is used as input to the generators. This is because we aim to learn a generative model that learns a distribution over goals rather than just a single goal. By sampling from the distribution, in expectation we can cover multiple high-regret regions. Furthermore, the same formulation of a latent noise variable input is also used in GoalGAN, the generative baseline we compare against. **We have added a sentence to the paper to clarify this design choice.**
>
> > In your method there is not really a "main solver" or agent, when reporting your results which one you pick? Why?
>
> Good question! With symmetrization,  Alice and Bob perform similarly in expectation, both in theory and practice. For consistency we report the results from Bob only.
>
> > You cite the recent work from Campero et al. 2021 about AMIGO which also uses a goal generator agent and a goal achiever agent, it could be interpreted as a 2-agent version of what is presented here but in discrete spaces, I missed then a more in depth comparison with their work in benchmarks or , at least, theoretically.
>
> **We have updated the related work section to discuss the differences to their work in more detail.** Outside of working in discrete spaces, one difficulty of applying their approach is the use of hyperparameters for the teacher to determine goal feasibility (e.g. how much the teacher is rewarded, heuristic scaling of time limit bounds to measure 'suitable effort'). The aim of this work is to blend the benefits of multiagent interactions that automatically give a measure of goal feasibility and a teacher reward without relying on hyperparameters or heuristics (eg. as in ASP, PAIRED) , with the benefits of explicit goal generation (eg. as in AMIGo, GoalGAN, etc.).

---

> > ### Author Response · Authors · 2021-11-16
> > **Response to A4MW (2/2)**
> >
> > > Since your work is not the first one exploring the idea of an agent looking for achievable yet challenging goals using a goal-generator agent and a reacher, the main contribution seems to be in the introduction of multiple of these and how they help each other, but then why stop in four? From the very beginning your work raises that question, thus is kind of disappointing that it is just left for future work.
> >
> > We note that the contributions of this work are not just in introducing a symmetric setup for the goal generation game, but also studying the training intricacies and design choices, as shown in Section 3 and Table 2.  Prior work (ASP, PAIRED) uses on policy methods, which are less sample efficient and prone to forgetting, and our contribution also consists of motivating using SAC for goal generation (beyond just sample efficiency, the replay buffer helps with memory and the entropy regularization helps with goal diversity) and handling the corresponding non-stationary challenges of using an off policy algorithm in a multiagent setting. **We have added a more in depth ablation of the specific components of our method in Appendix B.5.1 to highlight these aspects.**
> >
> > That said, we do agree that extending the approach to more agents would be an interesting case to study for future work. However, it is not trivial to construct a similar game with an increasing number of agents. For example, regret is defined as the difference between the returns of two agents. How can we extend this to multiple learning agents? One potential approach can be using the range in performance of multiple agents, although it is less clear if this is necessary and what gains this would provide, given the tradeoff of training more agents. Furthermore, if we introduce multiple goal generators, then we no longer have the structure of a zero-sum game between the generators, which can affect training stability and convergence. Careful consideration needs to be made to extend this method to multiple generators and/or learners. While interesting, these extensions are outside the scope of this work, and we hope that our investigations can inspire future work in this space!
> >
> > We hope we have addressed all the concerns from the reviewer. If there are any further questions, please let us know and we would be happy to answer.

---

> > > ### Comment · Reviewer_A4MW · 2021-11-19
> > > **Authors, please check the style file of the updated version**
> > >
> > > Authors, as I was reviewing the discussion and the updated version I saw that the updated submission has the NeurIPS submission format. I am warning you because this is probably fixable in a few minutes and there is time to update the submission with the ICLR format again (also note that the current title is not the one of the ICRL submission).

---

> > > > ### Author Response · Authors · 2021-11-19
> > > > **Updated Style**
> > > >
> > > > Thank you for letting us know, the document style should be fixed now.

---

> > > > > ### Comment · Reviewer_A4MW · 2021-11-19
> > > > > **Overlength**
> > > > >
> > > > > The style is fixed but the paper is 5 lines overlength now. I would suggest to fix this before the end of the rebuttal period.
> > > > >
> > > > > (Also, thank you for highlighting the changes, very useful!)

---

> > > ### Comment · Reviewer_A4MW · 2021-11-19
> > > **Some further points**
> > >
> > > First I would like to thank the authors for responding to our questions and concerns and the addition of new clarifications and experiments in the appendix that strength the submission.
> > >
> > > Some points I would suggest that authors further improve are:
> > >
> > > > We directly use the implementation from https://github.com/denisyarats/pytorch_sac, where the actor and critic are separated.
> > >
> > > I would include this information in the Appendix
> > >
> > > Figure 4, is not mentioned in the main text
> > >
> > > > With symmetrization, Alice and Bob perform similarly in expectation, both in theory and practice. For consistency we report the results from Bob only.
> > >
> > > I would suggest including that information in the main document
> > >
> > > Also, I am happy with the updates and the new ablations in B.5.1, they provide very good information and increase my confidence in the authors' work. I am concerned though, that too much of this material is in the Appendix and it is barely referred in the main document, specifically authors only refer to it with "For ablations on the components of our method, see Appendix B.5.1". I would strongly suggest to make space for 2/3 lines briefly summarising what ablations are done and the results they offer.

---

> > > > ### Author Response · Authors · 2021-11-19
> > > > **Addressing Comments**
> > > >
> > > > Thank you for the helpful suggestions for improvement, we have incorporated all suggestions to the latest draft. Please let us know if there are any further questions or recommendations!

---

> > > > > ### Comment · Reviewer_A4MW · 2021-11-20
> > > > > **Thank you and updated scores**
> > > > >
> > > > > I am glad that my suggestions helped. I checked the latest version and I believe that the work is in much better shape, and now supports better your work and contributions. I am updating my scores to recommend the acceptance of the paper.

---

> > > > > > ### Author Response · Authors · 2021-11-24
> > > > > > **Thank you**
> > > > > >
> > > > > > Thank you for increasing your score! We greatly appreciate the helpful suggestions which have improved the current draft.

---

### Official Review · Reviewer_JLa8 · 2021-11-02

**Correctness:** 4
**Technical Novelty And Significance:** 3
**Empirical Novelty And Significance:** 3
**Recommendation:** 6
**Confidence:** 4

**Main Review:**

Overall I found the paper written well and easy to understand. I thought the dynamic regret updates to the goal generating SAC buffer to be intuitive and appreciated the toy example they use to show it can better optimizing a non-stationary objective.

The main question for me that arose was: Where does the difference in Alice and Bob's performance come from? They both get to train on the same tasks and are the same algorithms. Is it from their different initializations? Is it from the fact that they first train on goals in different orders? Or is it simply that they are stochastic policies and when a task is somewhat hard one is likely to fail? If it is this last option, then I'd think you could come up with a similar and more compute efficient algorithm where a single generator is trying to pick goals to maximize the variance over returns of a single student. So in this setup you could have your single student just do 2 rollouts and use the difference in returns as the reward for the generator.

It feels like the H.E.R baseline should be included since you are using SAC to optimize your policies.

Why does the uniform goal distribution policy do so poorly on the reach task (Fig 7) with an extra goal dimension? I don't have a good intuitive sense for why CuSP should do better in this situation.

**Summary Of The Paper:**

The authors propose a new curriculum generating algorithm for goal conditioned agents. In their setup, they have two students and two teachers; each teacher's job is to find a task their student can do better than the other student. They find that this symmetric curriculum self play (CuSP) outperforms asymmetric goal generating algorithms such as asymmetric self play and goal gan in a variety of goal based simulated robotics tasks.

**Summary Of The Review:**

Overall I feel the paper is slightly above the acceptance threshold. It does a good job of explaining it's algorithmic choices with toy experiments and shows a reasonable bump over other curriculum methods on a series of goal conditioned tasks. However, I'm left with the question of why the method works at all and if there's a better formulation that achieves the same thing (see my first question in the main review). If this were clarified and some analysis included in the paper I think it would be a much stronger submission and would be a clear accept; in this case I would be more than happy to increase my score.

---

> ### Author Response · Authors · 2021-11-16
> **Response to JLa8**
>
> We thank the reviewer for their thoughtful comments and am glad they found the paper well written and easy to understand.
>
> We address specific comments below:
>
> > Where does the difference in Alice and Bob's performance come from? They both get to train on the same tasks and are the same algorithms. Is it from their different initializations? Is it from the fact that they first train on goals in different orders? Or is it simply that they are stochastic policies and when a task is somewhat hard one is likely to fail? If it is this last option, then I'd think you could come up with a similar and more compute efficient algorithm where a single generator is trying to pick goals to maximize the variance over returns of a single student. So in this setup you could have your single student just do 2 rollouts and use the difference in returns as the reward for the generator.
>
> Good question! We make use of all of the above -- Alice, Bob, and their respective goal generators are initialized randomly, in the symmetric setup Alice and Bob encounter different goal orders, and the policies are stochastic. The reviewer brings up an interesting question about using a single agent instead and looking at the variance in the rewards. **We investigate this case in Appendix B.5.1, Multiagent Motivation (L616), and find that only relying on stochasticity for the regret is insufficient.** We hypothesize that this is due to the single agent producing a much more flat regret landscape (i.e. most of the regrets are more similar) and the points of large regret may occur more by chance, making it challenging for the goal agent to identify a useful curriculum.
>
> >It feels like the H.E.R baseline should be included since you are using SAC to optimize your policies.
>
> Table 2 in Appendix B.2 points out environment specifics, where we are already using HER for the manipulation tasks as it is complementary to our method. For any environment with HER, we use it across all the ablations and baselines for a fair comparison. Although HER does induce a curriculum, we do not compare against HER alone as it does not explicitly generate goals that the agent has not encountered before, which is a different setting from CuSP.
>
> > Why does the uniform goal distribution policy do so poorly on the reach task (Fig 7) with an extra goal dimension? I don't have a good intuitive sense for why CuSP should do better in this situation.
>
> As goal dimensions increase, the uniform baseline becomes less likely to provide meaningful coverage over the goal space. In this misspecified goal dimension experiment, we add an additional dimension to the goal space of the goal generators which is not actually feasible in the environment and thus does not affect the agent return. That is, the effective goal space is only two dimensional. The naive uniform baseline will propose redundant goals in the extra third dimension as there is no way for the generator to learn or know that the third dimension is redundant. On the other hand, CuSP is designed to learn to find goals that maximize regret and over time, can figure out that the instantaneous regret is invariant to the generated value in the extra dimension and hence, explore more within the two feasible dimensions than in the redundant dimension. These experiments aim to show that in simpler goal spaces randomization alone may be sufficient, but in more complex goal spaces a learned approach is more appropriate since the goal generator can actively learn which parts of the goal space actually do or do not affect the regrets.
>
> We hope we have addressed all the concerns from the reviewer. If there are any further questions, please let us know and we would be happy to answer.

---

> > ### Author Response · Authors · 2021-11-21
> > **Follow Up**
> >
> > Please let us know if our response and updated draft have addressed the concerns you have raised. We are happy to address any further questions.

---

> > ### Author Response · Authors · 2021-11-28
> > **Follow up - 2**
> >
> > Thanks again for reviewing our paper! Since we are nearing the end of the discussion period (tomorrow), we were wondering if our response and updated draft addressed your comments. We are happy to address any further questions in the remaining time.

---

### Official Review · Reviewer_qGDC · 2021-11-03

**Correctness:** 3
**Technical Novelty And Significance:** 2
**Empirical Novelty And Significance:** 2
**Recommendation:** 5
**Confidence:** 3

**Main Review:**

The authors say this is a 2-player zero-sum game but (1) they do not solve this game i.e. find a Nash equilibrium. Self play does not converge to a Nash equilibrium. (2) there exist Nash equilibria of this game where Alice and Bob are identical in which case the two goal generators get zero reward no matter what goals they propose. (3) even if one solved a Nash of this game that wasn’t degenerate, it’s not clear to me why that would be a good curriculum.
So it seems that the method mainly relies on the training dynamics, not the static equilibrium properties of this setup. But even then, let’s say Alice and Bob start out as identical and the two goal proposers also start out as identical. Then shouldn’t Alice and Bob remain identical and the two goal proposers also remain identical because they all have the same updates?
I don’t see how the given method satisfies the desiderata (progressive diversity, progressive feasibility and anti-forgetting). It seems like the first three rows are not a core part of the method but are used to heuristically improve the results. The last row refers to the actual newly proposed objective where you have symmetric goal setters and solvers. But as mentioned above, I don’t think this symmetric objective gives the desiderata like claimed.
Why is SAC used to propose goals? Isn’t a goal a single action?
The results look decent, but do not perform much better than uniform (domain randomization) on most tasks. I would be interested to see how well PAIRED does on these tasks.


**Summary Of The Paper:**

This paper proposes a modification to asymmetric self play where we have two goal generators and two solvers. Each goal generator is incentivized to propose goals that one solver can solve but the other cannot and each solver is incentivized to solve goals given to them. In experimental results the method seems to perform a bit better than domain randomization.


**Summary Of The Review:**

I like this direction but I am worried that the main objective has problems that can prevent it from being an effective curriculum, even if they don't show up in these particular results.

---

> ### Author Response · Authors · 2021-11-16
> **Response to qGDC (1/2)**
>
> We thank the reviewer for their thoughtful comments and am glad they like the direction of the paper.
>
> We address specific comments below:
>
> >The authors say this is a 2-player zero-sum game but (1) they do not solve this game i.e. find a Nash equilibrium. Self play does not converge to a Nash equilibrium.
>
> >(3) even if one solved a Nash of this game that wasn’t degenerate, it’s not clear to me why that would be a good curriculum.
>
> We agree with the reviewer that we do not explicitly aim to solve for Nash equilibrium, but rather use the training dynamics for automatic curricula generation. The reviewer brings up the point that self play does not converge to a Nash equilibrium, however in this work we are not using self play. The learners and goal generators are all different agents with their own policy updates.
>
> > (2) there exist Nash equilibria of this game where Alice and Bob are identical in which case the two goal generators get zero reward no matter what goals they propose.
>
> > But even then, let’s say Alice and Bob start out as identical and the two goal proposers also start out as identical. Then shouldn’t Alice and Bob remain identical and the two goal proposers also remain identical because they all have the same updates?
>
> The reviewer brings up a good point that there can be equilibria where Alice and Bob are identical, which we touch on in Note 2 of Appendix C. Here, we note that in such an initialization, Alice and Bob are either already capable of achieving all goals (and thus are done learning), or if there are still goals that neither Alice and Bob can achieve, then the exploratory components of the goal generator (with entropy regularized SAC, or making the goal generators $\epsilon$-greedy policies) will eventually propose such goals as well, even though they have zero regret. Furthermore, as the learner policies are stochastic, this makes it highly unlikely that all agents will remain identical throughout learning. We also empirically investigate conditions with flat regret landscapes (i.e. Bob and Alice are mostly identical) in the toy experiments in Figures 2 and 3, which motivated our method design choices.
>
>
> > I don’t see how the given method satisfies the desiderata (progressive diversity, progressive feasibility and anti-forgetting).It seems like the first three rows are not a core part of the method but are used to heuristically improve the results. The last row refers to the actual newly proposed objective where you have symmetric goal setters and solvers. But as mentioned above, I don’t think this symmetric objective gives the desiderata like claimed.
>
> > I would be interested to see how well PAIRED does on these tasks.
>
> **To validate that the first three rows are a core part of the method, we include additional experiments in Appendix B.5.1 ablating the first three components to emphasize their importance to the method and compare the base version to PAIRED** (which can be seen as an ablation of CuSP without symmetrization, entropy regularization in the generator, regret replay buffers, and dynamic regret updates). We also experimented with using PPO for the goal generator as in PAIRED, but found this was too sample inefficient to perform well on our evaluation tasks and the toy experiment in Figure 2 also shows PPO is less capable of handling the flat regret landscape. **In Figure 9, we find that removing the first three rows of Table 1 greatly degrades performance, and qualitatively we can see that removing those components leads to worse goal generation in Figure 10.** The components of the method are indeed empirically validated, however, they were carefully chosen and we describe throughout Section 3 the intuition behind each component and how they connect to the desiderata. We hope this more detailed ablation emphasizes that each component is a core part of the method, and are happy to answer any further questions.

---

> > ### Author Response · Authors · 2021-11-16
> > **Response to qGDC (2/2)**
> >
> >
> > > Why is SAC used to propose goals? Isn’t a goal a single action?
> >
> > Good question!  As noted in the introduction, we are motivated to create curricula that balance progressive exploration with anti-catastrophic exploitation. This lends itself naturally to RL objectives, which are designed around the exploration and exploitation dilemma. As we note in the paper (L145), goal proposal is a single action and while single-step RL may be an uncommon usage for SAC, there has been prior work in one-step RL. To justify the choice of SAC for goal proposal, we empirically investigate three different optimization approaches for the goal generator in Figure 2 -- Adam, PPO, and SAC -- in a toy task that represents a challenging scenario for the goal generator. As the reviewer noted earlier, similar performance between Alice and Bob can be problematic, so we constructed the toy task to consist of a mostly flat landscape. We find that for such a flat landscape, SAC is much better at finding the optima than the other optimization approaches.
> >
> > > The results look decent, but do not perform much better than uniform (domain randomization) on most tasks.
> >
> > While domain randomization does well on the random evaluations in Figure 5, we would like to emphasize that on the harder tasks in Figure 6, the gap between our method and domain randomization is larger. In Behind Obstacles, our method converges to near 100% success about 5x faster, in Walker Far Right only our method is able to achieve non-zero success, and in Toss Upwards our method achieves 80% success while domain randomization achieves 30% success. Furthermore, in the misspecified goal dimension case in Figure 7, we find that our method converges to 80% success and 70% success for in distribution and out of distribution goals respectively while domain randomization converges to 30% success for both.  We note that domain randomization is particularly strong for the random OOD evaluations in Figure 5 as these evaluations assess ability to achieve random goals, which the domain randomization baseline is well suited for. Even so, our method matches or outperforms the domain randomization baseline.
> >
> > We hope we have addressed all the concerns from the reviewer. If there are any further questions, please let us know and we would be happy to answer.

---

> > > ### Author Response · Authors · 2021-11-21
> > > **Follow Up**
> > >
> > > Please let us know if our response and updated draft have addressed the concerns you have raised. We are happy to address any further questions.

---

> > > > ### Comment · Reviewer_qGDC · 2021-11-22
> > > > **Response**
> > > >
> > > > Thank you for adding ablations, I have bumped my score up to a 5 but still don't recommend acceptance because of the possibility of convergence to bad equilibria and the fact that the method is a set of four small changes to PAIRED.

---

> > > > > ### Author Response · Authors · 2021-11-24
> > > > > **Thank you**
> > > > >
> > > > > Thank you for increasing your score! We appreciated the insightful questions and they have helped us to improve the current draft. We would like to emphasize that while our method does build on top of the regret objective proposed in PAIRED, the additions we propose are crucial for these continuous control tasks. As we see in Figure 9 in the updated Appendix, solely using the PAIRED objective plateaus slightly above 20% success for the random OOD evaluations in Toss, while adding all the components of CuSP leads to 80% success.

---

### Official Review · Reviewer_GBqb · 2021-11-03

**Correctness:** 4
**Technical Novelty And Significance:** 3
**Empirical Novelty And Significance:** 3
**Recommendation:** 6
**Confidence:** 3

**Details Of Ethics Concerns:**

I don't have ethics concerns for this paper.

**Main Review:**

Strength:
1. The idea is well-motivated and clearly presented
2. Comprehensive experiments are performed to verify the effectiveness of the proposed method

Weakness:
1. The curriculum generation is solely based on the concept of goals. The method would be more powerful if it can be used for non-goals based settings, such as those discussed in [1]. If would be better if the authors can provide some discussion about the general applicability of the proposed method.

2. The work is mostly based on empirical study, the paper could be strengthened if some theoretical analysis, such as sample/computation complexity, convergence etc. can be provided. Since this method needs to consider multiple agent learning, the computation overhead could be a concern.

Other comments:
1. For “Goal Generator Design”, It is unclear to me why a latent noise variable z need to be introduced.

2. For the objective of goal generator policy, how is the exploitation term be constructed, especially under one-step optimization, does it include all the immediate rewards encountered in one episode?

[1] Portelas et al., Automatic Curriculum Learning for Deep RL: A Short Survey.


**Summary Of The Paper:**

This paper proposed an automated goal generation method, Curriculum selfplay (CuSP) which aims to generate goals of diversity and of increasing difficulty in sample efficient way, as well as to prevent catastrophically forgetting previously solved goals. The method is based on a 4-player game formulation, where cooperation and competition are balanced between two off-policy student learners and two regret-maximizing teachers, who plays the role of balancing progressive exploration and anti-catastrophic forgetting. The method is evaluated on a arrange of continuous control tasks and is shown to outperform a few baselines.

**Summary Of The Review:**

In all, this paper is well written and provided a novel formulation for goal-conditional automatic curriculum learning. The paper could be strengthened if its use cases can be extended to more general curriculum learning settings (beyond goal-generation). It would also be better if more theoretical analysis, such as sample complexity/convergence and be provided.

---

> ### Author Response · Authors · 2021-11-16
> **Response to GBqb**
>
> We thank the reviewer for their thoughtful comments and am glad they found the paper well motivated with comprehensive evaluations.
>
> To address specific comments:
>
> > The method would be more powerful if it can be used for non-goals based settings, such as those discussed in [1]. If would be better if the authors can provide some discussion about the general applicability of the proposed method.
>
> As noted in [1], Automatic Curriculum Learning (ACL) methods can be considered to fall under a few categories of task MDP control -- goals, environments, reward functions, etc. [1] cites many goal-specific ACL methods, two of which we use as baselines in this work. Goal generation control is one of the main techniques in ACL for multi-goal reinforcement learning, and the scope of this work is specifically focused on this category as we aim to tackle curriculum generation for goal-conditioned RL. **We have updated the paper to reference the survey and make this categorization more clear.** We also note that CuSP’s algorithm can be extended to other goal formulations (e.g. language), which would be exciting future work.
>
> > The paper could be strengthened if some theoretical analysis, such as sample/computation complexity, convergence etc. can be provided. Since this method needs to consider multiple agent learning, the computation overhead could be a concern.
>
> For a game theoretic analysis of the CuSP framework, see Appendix C. The reviewer makes a good point that a theoretical analysis of complexity can help with comparing multiagent methods, as they may be more computationally intensive. To this point, we note that a thorough complexity analysis in this line of work (multi-agent RL) is challenging with many open problems, and prior curriculum generation methods do not provide any such guarantees. That said, we would like to note that our method is formulated in such a way that the rounds can be fully parallized -- i.e. since the agents do not interact directly in the environment, the goal generators can propose goals simultaneously and Alice and Bob can carry out rollouts and act simultaneously as well. **We also conduct additional experiments in Appendix B.5.1 Multiagent Motivation, with a single learner agent (as suggested by Reviewer JLa8), and find that the performance decreases greatly.** This indicates that the multiagent aspect is necessary for successful curriculum generation.
>
> >For “Goal Generator Design”, It is unclear to me why a latent noise variable z need to be introduced.
>
> This design choice builds on standard practice in generative models (e.g. GANs, VAEs) where a latent noise variable is used as input to the generators. This is because we aim to learn a generative model that learns a distribution over goals rather than just a single goal. By sampling from the distribution, in expectation we can cover multiple high-regret regions. Furthermore, the same formulation of a latent noise variable input is also used in GoalGAN, the generative baseline we compare against. **We have added a sentence to the paper to clarify this design choice.**
>
> > How is the exploitation term be constructed, especially under one-step optimization, does it include all the immediate rewards encountered in one episode?
>
> The exploitation term, $\mathfrak{R}^G(s,g)$ is the regret for a particular goal, computed after both $\pi_A$ and $\pi_B$ rollout their policies conditioned on $g$ and starting at $s$. Explicitly we have $\mathfrak{R}^G(s,g) = \sum_{t=1}^T \gamma^t (R^A(s_t, g_t) - R^B(s_t, g_t))  $ where $R^A$ and $R^B$ are the single step rewards attained by $\pi_A$ and $\pi_B$ respectively, discounted by $\gamma$. Since the regret exploitation term is the difference in the discounted sum of rewards for the agents, it does indeed include all the intermediate rewards encountered in an episode.
>
> We hope we have addressed all the concerns from the reviewer. If there are any further questions, please let us know and we would be happy to answer.

---

> > ### Author Response · Authors · 2021-11-21
> > **Follow Up**
> >
> > Please let us know if our response and updated draft have addressed the concerns you have raised. We are happy to address any further questions.

---

> > ### Author Response · Authors · 2021-11-28
> > **Follow up - 2**
> >
> > Thanks again for reviewing our paper! Since we are nearing the end of the discussion period (tomorrow), we were wondering if our response and updated draft addressed your comments. We are happy to address any further questions in the remaining time.

---

### Author Response · Authors · 2021-11-29
**Summary of Changes**

We thank the reviewers again for their thorough feedback and reviews. The following is a summary of changes made to address reviewer concerns and suggestions, which are highlighted in pink text in the PDF:

Main Text:
- We added a citation to *Automatic Curriculum Learning for Deep RL: A short survey.* (Portelas et al., 2020) with a categorization for our approach.
- We clarified the motivation behind sampling from a random noise variable for the goal generator.
- We conducted an additional detailed ablation of the components of our method listed in Table 1 in Appendix B.5.1, with a more direct comparison to PAIRED, and find that each component is necessary for improving task performance.
- We provide more details on the goal space specification of the environments.
- We provide more discussion on a comparison to AMIGo.

Appendix:
- As mentioned in the main text, we give details on the environment goal space specification.
- As mentioned in the main text, we ran additional ablations on each individual component of our method and find that the entropy regularization and replay buffer are necessary, and that symmetrization alone with the regret objective is not sufficient.
- As suggested by Reviewer JLa8, we ran additional experiments with a single agent using only variance over returns for the regret. We find that this leads to much worse task performance, and hypothesize that variance alone is insufficient for identifying a useful curriculum (i.e. the regret landscape is likely to be more flat with similar returns when only using a single agent, which makes it harder for the goal generator).

We would be happy to address any remaining questions.

---

### Decision · Program_Chairs · 2022-01-20

**Decision:**

Accept (Poster)

**Comment:**

While one reviewer remained concerned about the possibility of convergence to bad equilibria and felt that the proposed method appears to be four minor changes from prior work (PAIRED), the authors demonstrate empirically that the proposed changes make a significant difference in their evaluation. Other reviewers were positive about this work and all others rated this work as an accept. Post rebuttal the most positive reviewer increased their score to an 8 and felt did a good job answering their concerns. They wanted to see an analysis of systems with larger numbers of agents, but felt that the current manuscript was more than sufficient to warrant acceptance, and fell into the category of a good paper with the additional ablations provided during the rebuttal.

The AC recommends accepting this paper.